# DISENTANGLED HETEROGENEOUS COLLABORATIVE FILTERING

## ABSTRACT

Modern recommender systems often utilize low-dimensional latent representations to embed users and items based on their observed interactions. However, many existing recommendation models are primarily designed for coarse-grained and homogeneous interactions, which limits their effectiveness in two key dimensions: i) They fail to exploit the relational dependencies across different types of user behaviors, such as page views, add-to-favorites, and purchases. ii) They struggle to encode the fine-grained latent factors that drive user interaction patterns. In this study, we introduce DHCF, an efficient and effective contrastive learning recommendation model that effectively disentangles users' multi-behavior interaction patterns and the latent intent factors behind each behavior. Our model achieves this through the integration of intent disentanglement and multi-behavior modeling using a parameterized heterogeneous hypergraph architecture. Additionally, we propose a novel contrastive learning paradigm that adaptively explores the benefits of multi-behavior contrastive self-supervised augmentation, thereby improving the model's robustness against data sparsity. Through extensive experiments conducted on three public datasets, we demonstrate the effectiveness of DHCF, which significantly outperforms various strong baselines with competitive efficiency.

## 1 INTRODUCTION

Modern recommender systems aim to learn the interests of users from the observed user-item interactions (*e.g.*, click or purchase), with the collaborative filtering (CF) frameworks for making item recommendations (Chen et al., 2020b). Most existing CF paradigms are designed to project users and items into latent embedding space to be reflective of users' preference through various learning techniques (Wu et al., 2021b; Yang et al., 2022a), *e.g.*, MLP (He et al., 2017), Auto-encoder (Sedhain et al., 2015) and graph neural networks (GNNs) (Wang et al., 2019; 2020b; Fan et al., 2022).

Recent studies mainly focus on learning disentangled representations behind a single type of user-item interaction, without taking behavior heterogeneity into consideration. Examples of such studies include the variational auto-encoder MacridVAE (Ma et al., 2019b), GNN-based model DGCF (Wang et al., 2020a), and geometry-based recommender GDCF (Zhang et al., 2022). Although there are potential benefits to jointly performing intent disentanglement and multi-behavior modeling for recommendation, the problem of disentangled heterogeneous collaborative filtering is still underexplored. This is due to two key challenges. Firstly, the intent disentanglement should be carefully designed to ensure that the factorized representations reflect the expressive information pertinent to different types of user-item interactions. Secondly, due to the limited amount of target behavior data (*e.g.*, purchase, like), most current multi-behavior recommenders (*e.g.*, EHCF (Chen et al., 2020a), MGCN (Jin et al., 2020), KHGT (Xia et al., 2021b)) may fall short in effectively customizing knowledge transfer between the auxiliary and target behaviors. Therefore, self-supervised tasks need to be well-designed for disentangled contrastive learning on heterogeneous user-item interactions in an adaptive manner.

In light of the above challenges, we propose a new hypergraph contrastive learning model called DHCF, which addresses behavior heterogeneity and disentangled intent factors through adaptive multi-behavior data augmentation. DHCF first aims to encode behavior-aware latent intents of users using a parameterized heterogeneous hypergraph, generating factorized representations that are specific to each type of user-item interaction. To achieve this, DHCF employs tailored hyperedges to aggregate intent embeddings, leveraging disentangled global collaborative relationships. To enhance

the generalization capacity for encoding personalized multi-behavior dependencies from sparse interaction data, we introduce a behavior-wise contrastive learning paradigm that models node- and graph-level behavior heterogeneity using adaptive self-supervision. Through this approach, DHCF incorporates multi-behavior discrimination objectives into adaptive contrastive learning, ensuring that the factorized representations with intent disentanglement accurately reflect the heterogeneous collaborative context. We release our model implementation with source code and evaluated datasets at the link: `https://anonymous.4open.science/r/DHCF-main-2658/`.

- We devise a new heterogeneous collaborative filtering model DHCF, which uncovers latent intent associated with multiplex user-item interactions, thereby enhancing recommendation outcomes.
- We propose a behavior-wise contrastive learning model that facilitates adaptive data augmentation at both the node and graph levels. This paradigm enables the simultaneous extraction of local and global self-supervised collaborative signals, effectively integrating relation heterogeneity encoding with intent disentanglement, to achieve more expressive and robust user preference learning.
- We provide theoretical analyses to demonstrate how our hypergraph-based representation disentanglement enriches and enhances the behavior-wise contrastive learning by incorporating additional and adaptive supervision signals associated with the extracted global connectivity.
- We evaluate our DHCF on three public recommendation datasets and demonstrate significant performance improvements compared to 18 baselines from 6 research lines. Additionally, we conduct thorough analysis to validate the robustness, efficiency, and interpretability of our model.

## 2 RELATED WORK

**Multi-Behavior Recommender Systems**. Recent studies have proposed multi-behavior recommenders that leverage different types of user interactions to improve predictions on target behaviors (Chen et al., 2021a; Meng et al., 2023). Examples of such systems include the graph convolution-based method MBGCN (Jin et al., 2020), the graph attention-based approach KHGT (Xia et al., 2021b), and the memory-based attentive model GNMR (Xia et al., 2021a). SMRec (Gu et al., 2022) involves a star-type contrastive regularization to generate self-supervised signals for modeling behavior differences. Xuan et al. (2023) proposes a contrastive learning method for knowledge-enhanced multi-behavior recommender. However, none of these studies investigate the benefits of learning diverse latent intent factors behind heterogeneous user-item interactions. In this work, we address this challenge with the parameterized heterogeneous hypergraph network with multi-channel space.

**Disentangled Representation for Recommendation**. The goal of disentangled representation learning-enhanced recommendation lies in disentangling the latent factors beneath the complex interaction data, based on various techniques (Chen et al., 2021b). For example, DGCF (Ma et al., 2019a) is built over the GCN to perform intent-aware disentangled message passing. MacridVAE (Ma et al., 2019b) aims to disentangle user representations using variational auto-encoder. Inspired by the representation learning over hybrid geometries, GDCF (Zhang et al., 2022) disentangles user-item interactions with non-Euclidean anatomy. Wu et al. (2022) proposes a disentangled contrastive learning approach for social recommendation. Existing solutions do not take the heterogeneous user behaviors into consideration, which present unique challenges for disentangled collaborative filtering.

**Contrastive Learning for Recommendation**. Contrastive learning (CL) has been widely adopted in various recommendation tasks, including general collaborative filtering based on graph models, and sequential recommendation based on sequence models. For example, SGL (Wu et al., 2021a) and SimGCL (Yu et al., 2022) enhance the graph neural collaborative filtering paradigms by reinforcing user and item representations with the contrastive self-discrimination task. CL4SRec (Qiu et al., 2022) and DuoRec (Xie et al., 2022) propose to augment sequence representations using masking, cropping, and reorder operations. Additionally, CL techniques have been used in other applications such as micro-video recommendation (Yi et al., 2022; Wei et al., 2021), knowledge graph-enhanced recommendation (Yang et al., 2022b), and cross-domain recommendation (Xie et al., 2022).

## 3 METHODOLOGY

### 3.1 PRELIMINARIES

In our heterogeneous collaborative filtering scenario, the relationships between users ($u_i$ $i \in [1, ..., I]$) and items ($v_j$ $j \in [1, ..., J]$) are often exhibited with diverse interactions (*e.g.*, click, review, purchase).

In general, $K$ types of heterogeneous user-item interactions are partitioned into the target behavior (*e.g.*, purchase in e-commerce, or like in video streaming platform), and the other auxiliary behaviors, such as {click, add-to-favorite} and {watch, review}. We define a three-way tensor $\mathcal{X} \in \mathbb{R}^{I \times J \times K}$ to represent the heterogeneous interaction data. Specifically, each element $x_{i,j,k} = 1$ indicates that user $u_i$ has interacted with item $v_j$ with the $k$-th behavior type and $x_{i,j,k} = 0$ for unobserved interactions.

**Problem Statement**. With the foregoing definitions, the heterogeneous collaborative filtering for multi-behavior recommendation can be formally stated as follows: **Input**. The observed user-item heterogeneous interactions $\mathcal{X}$. **Output**. The learning function $f(\cdot)$ that makes predictions on the unobserved user-item interactions with the target interaction type (*e.g.*, purchase, like).

## 3.2 MULTIPLEX GRAPH RELATION LEARNING

Given the heterogeneous interaction data $\mathcal{X}$, we generate a multiplex interaction graph $\mathcal{G}$ to reflect diverse relationships among users and items. Users and items construct the set of nodes in $\mathcal{G}$. In our heterogeneous collaborative filtering scenario, multiplex edges commonly exist between each user-item pair to indicate different types of interaction behaviors. Inspired by the success of GNN-based CF (Zhang et al., 2019; Yu & Qin, 2020), we design our relation-aware message passing as follows:

$$\mathbf{z}_{i,k}^{(u)} = \sum_{j \in \mathcal{N}_{i,k}} m_{i,j} \circ \mathbf{e}_j, \quad \mathbf{z}_{j,k}^{(v)} = \sum_{i \in \mathcal{N}_{j,k}} m_{j,i} \circ \mathbf{e}_i, \quad \bar{\mathbf{z}}_i^{(u)} = \sum_{k=1}^{K} \mathbf{z}_{i,k}^{(u)}, \quad \bar{\mathbf{z}}_j^{(v)} = \sum_{k=1}^{K} \mathbf{z}_{j,k}^{(v)} \quad (1)$$

where $\mathcal{N}_{i,k}$ and $\mathcal{N}_{j,k}$ denotes the set of neighboring nodes for user $u_i$ and item $v_j$ under the $k$-th type of behavior, respectively. To alleviate the overfitting effect, we apply $m_{i,j}, m_{j,i} \in \{0, 1\}$ as the dropout operator with the binary value during the information aggregation (Wu et al., 2021a). $\circ$ denotes the broadcasting multiplication operation. $\mathbf{e}_i, \mathbf{e}_j \in \mathbb{R}^d$ denote the general node embeddings for $u_i$ and $v_j$, initialized by random sampling. $\mathbf{z}_{i,k}^{(u)}, \mathbf{z}_{j,k}^{(v)} \in \mathbb{R}^d$ are the type-specific behavior embeddings for $u_i$ and $v_j$ with behavior type $k$. We generate the aggregated multi-typed behavior representation $\bar{\mathbf{z}}_i^{(u)}, \bar{\mathbf{z}}_j^{(v)} \in \mathbb{R}^d$ using the sum-pooling operator. To improve model efficiency (He et al., 2020; Chen et al., 2020b), DHCF provides the compact design of graph-based message passing.

## 3.3 RELATION-AWARE INTENT DISENTANGLEMENT

To learn the diverse intentions behind heterogeneous user preference in recommendation, we propose to consider the disentanglement of latent factors with the behavior heterogeneity over different types of user-item interactions. Inspired by the strength of hypergraph structure for modeling high-order connectivity (Feng et al., 2019; Yi & Park, 2020), we construct our disentangled heterogeneous hypergraph to integrate the diversity of user intents with the multi-behavior user-item interactions. Specifically, for each type of user-item interactions, we generate multi-channel hyperedges with size $E$ representing the number of latent intents for user interaction preference. Each hyperedge connects different user/item nodes through the parameterized node-hyperedge dependency modeling, to capture the relation-aware global collaborative relationships. In form, the learnable hypergraph adjacent matrices $\mathcal{H}_k^{(u)} \in \mathbb{R}^{I \times E}, \mathcal{H}_k^{(v)} \in \mathbb{R}^{J \times E}$ connecting nodes and hyperedges under the $k$-th type of user-item interactions are generated as follows:

$$\mathcal{H}_k^{(u)} = \mathbf{Z}_k^{(u)} \cdot \mathbf{W}_k^{(u)\top}, \quad \mathcal{H}_k^{(v)} = \mathbf{Z}_k^{(v)} \cdot \mathbf{W}_k^{(v)\top} \quad (2)$$

where $\mathbf{E}_k^{(u)} \in \mathbb{R}^{I \times d}, \mathbf{E}_k^{(v)} \in \mathbb{R}^{J \times d}$ are behavior-aware user and item embeddings composed by the aforementioned $\bar{\mathbf{z}}_i^{(u)}, \bar{\mathbf{z}}_j^{(v)}$. Parameters $\mathbf{W}_k^{(u)}, \mathbf{W}_k^{(v)} \in \mathbb{R}^{E \times d}$ represent the embedding matrices for hyperedges related to the $k$-th interaction type for users and items, respectively. Each hyperedge embedding vector encodes the latent features for a specific user intent.

## 3.4 PARAMETERIZED MULTI-BEHAVIOR HYPERGRAPH

With the learnable hypergraph structures for latent intent disentanglement, DHCF performs hypergraph-guided message passing to refine user and item embeddings, via modeling of user- and item-wise global collaborative relations with fine-grained intent levels. We define our behavior-aware

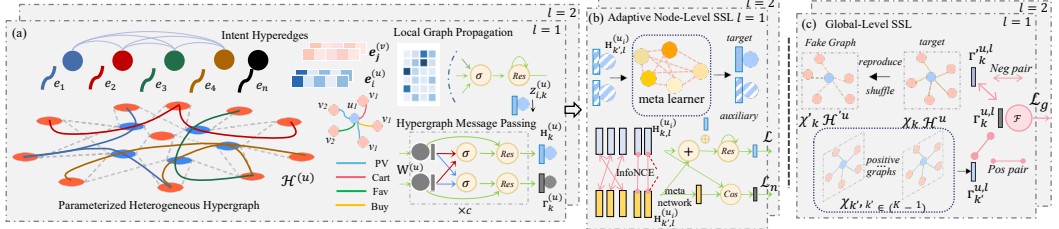

Figure 1: (a) Relation-aware intent disengagement is enabled with our designed parameterized heterogeneous hypergraph for generating disentangled representation with the behavior heterogeneity preserved. (b) Node-level contrastive learning injects self-supervised local collaborative signals with the adaptive cross-behavior knowledge transfer. (c) Graph-level contrastive learning paradigm performs data augmentation via the multi-behavior user self-discrimination with global context.

hypergraph embedding propagation between leaf nodes and intent hyperedges as follows:

$$\mathbf{H}_k^{(u)} = \delta(\tilde{\mathcal{H}}^{(u)} \cdot \mathbf{\Gamma}_k^{(u)}) = \delta(\tilde{\mathcal{H}}^{(u)} \cdot \delta(\tilde{\mathcal{H}}^{(u)\top} \cdot \mathbf{Z}_k^{(u)}))$$
$$\mathbf{H}_k^{(v)} = \delta(\tilde{\mathcal{H}}^{(v)} \cdot \mathbf{\Gamma}_k^{(v)}) = \delta(\tilde{\mathcal{H}}^{(v)} \cdot \delta(\tilde{\mathcal{H}}^{(v)\top} \cdot \mathbf{Z}_k^{(v)})) \tag{3}$$

where $\mathbf{H}_k^{(u)} \in \mathbb{R}^{I \times d}, \mathbf{H}_k^{(v)} \in \mathbb{R}^{J \times d}$ are the smoothed node embedding matrices under the $k$-th behavior type for users and items, respectively. $\delta(\cdot)$ denotes the non-linear activation function LeakyReLU. $\mathbf{\Gamma}_k^{(u)}, \mathbf{\Gamma}_k^{(v)} \in \mathbb{R}^{E \times d}$ denote the embeddings of the $E$ hyperedges under the $k$-th behavior type for users and items, respectively. $\mathbf{\Gamma}_k = \delta(\mathcal{H}^\top \mathbf{Z}_k)$. In our disentangled hypergraph module, DHCF takes $\mathbf{Z}_k^{(u)} \in \mathbb{R}^{I \times d}, \mathbf{Z}_k^{(v)} \in \mathbb{R}^{J \times d}$ (encoded from Equation 1) as the input node embeddings. Through behavior-aware hypergraph message passing with intent disentanglement, DHCF characterizes the diverse intents behind multiplex user-item interactions, leading to better user representations. With the heterogeneous embeddings $\mathbf{H}_k$, we generate overall user/item embeddings $\bar{\mathbf{H}}$ using sum-pooling.

## 3.5 HIGH-ORDER EMBEDDING PROPAGATION

By integrating the local graph convolutions and global hypergraph message passing, DHCF alternatively applies the two modules and recursively refines user/item embeddings for injecting high-order information. Taking the initialized node embeddings $\mathbf{E}^{(u)}, \mathbf{E}^{(v)}$ as the 0-order embeddings $\mathbf{\Lambda}^{(u,0)}, \mathbf{\Lambda}^{(v,0)}$, the $l$-th order node embeddings can be calculated iteratively as follows:

$$\mathbf{\Lambda}^{(u,l)} = \bar{\mathbf{Z}}^{(u,l)} + \bar{\mathbf{H}}^{(u,l)} + \mathbf{\Lambda}^{(u,l-1)}, \quad \bar{\mathbf{Z}}^{(u,l)} = f(\mathbf{\Lambda}^{(v,l-1)}), \quad \bar{\mathbf{H}}^{(u,l)} = g(\bar{\mathbf{Z}}^{(u,l)}) \tag{4}$$

where $f(\cdot)$ denotes the behavior-aware graph convolutions performing on the multiplex interaction graph $\mathcal{G}$. $g(\cdot)$ is the parameterized heterogeneous hypergraph neural network for relation-aware intent disentanglement. We further apply a residual connection (He et al., 2016) to alleviate the gradient vanishing issue. The final embeddings are generated through a multi-order aggregation as follows:

$$\mathbf{\Psi}^{(u)} = \sum_{l=0}^{L} \mathbf{\Lambda}^{(u,l)}, \quad \mathbf{\Psi}^{(v)} = \sum_{l=0}^{L} \mathbf{\Lambda}^{(v,l)} \tag{5}$$

## 3.6 MULTI-RELATION ADAPTIVE CONTRASTIVE LEARNING

The data sparsity issue is ubiquitous in heterogeneous collaborative filtering scenarios, *i.e.*, the majority of users have limited target behaviors (*e.g.*, purchase, like) as supervision labels. To enhance the model robustness for encoding heterogeneous collaborative relations, we design a hierarchical contrastive learning paradigm with the hypergraph structures, so as to derive self-supervised signals from the original multi-behavior interaction data. Towards this end, a multi-behavior hierarchical contrastive learning paradigm is proposed to holistically generate both node-level and graph-level contrastive objectives, while still maintaining the dependencies among different behavior types.

**Node-Level Adaptive Contrastive Augmentation**. Dependencies among different types of behavior may vary by users, due to their diverse interactive patterns in online platforms. For example, someone

are more preferable to favorite or like a video in TikTok while it is not easy to get likes on the watched videos from other persons. To capture such personalized multi-behavior patterns, we integrate the meta network encoder with the self-supervised learning for adaptive data augmentation. In our DHCF, cross-type behavior relationships are modeled through the multi-behavior self-discrimination with the preservation of personalized behavior semantics. We design our meta network encoder to transform different types of auxiliary behavior embeddings in an adaptive way as follows:

$$\widetilde{\mathbf{H}}_{i,k'}^{(u,l)} = \mathbf{H}_{i,k'}^{(u,l)} \cdot \delta(Norm(\mathbf{H}_{i,k'}^{(u,l)})\mathbf{W}_{i,k'}^{(u,l)} + \mathbf{b}_{i,k'}^{(u,l)}) \tag{6}$$

where $\widetilde{\mathbf{H}}_{i,k}^{(u,l)} \in \mathbb{R}^{I \times d}$ behavior-aware user embeddings which are adaptively transformed from the meta network for the $l$-th graph propagation layer. $\delta(\cdot)$ and $Norm(\cdot)$ denote the activation and the $l_2$ normalization function, respectively. $\mathbf{b}_{i,k'}^{(u,l)} \in \mathbb{R}^d$ and $\mathbf{W}_{i,k'}^{(u,l)} \cdot \in \mathbb{R}^{d \times d}$ are the trainable parameters corresponding to the $k'$-th type of auxiliary behaviors.

**Multi-Behavior User Self-Discrimination**. Our new contrastive learning paradigm aims to offer additional supervision signals, by reinforcing user representations via multi-behavior self-discrimination. Particularly, based on the multi-view behavior embeddings of users, the target and each type of auxiliary behavior of the same user are treated as positive pairs (*i.e.*, $(\widetilde{\mathbf{h}}_{i,k}, \widetilde{\mathbf{h}}_{i,k'})$). The embeddings of different users are considered as the negative samples (*i.e.*, $(\widetilde{\mathbf{h}}_{i,k'}, \widetilde{\mathbf{h}}_{i',k'})$). Here, $k'$ denotes the target behavior and $k$ denotes one auxiliary behavior. In our CL paradigm, the augmented self-supervised signals enforces the behavior divergence of different users. Following the adopted contrastive loss in Wu et al. (2021a), InfoNCE is applied to reach the node-level embedding agreement as follows:

$$\mathcal{L}_n = \sum_{i=1}^{I} \sum_{k=1}^{K} -\log \frac{\exp(s(\widetilde{\mathbf{H}}_{i,k'}, \widetilde{\mathbf{H}}_{i,k})/\tau)}{\sum_{i'=1}^{I} \exp(s(\widetilde{\mathbf{H}}_{i',k'}, \widetilde{\mathbf{H}}_{i,k'})/\tau)} \tag{7}$$

The node-level contrastive loss for the item dimension can be calculated in an analogous way. $s(\cdot)$ denotes the cosine function, which measures similarity after normalizing the input vectors. $\tau$ represents a temperature variable, which helps learn hard samples in self-supervised learning. The augmented self-supervision with our node-level adaptive contrastive learning, endows our model to jointly capture the commonality and divergence of multi-behavior preference.

**Graph-Level Multi-Relational Contrastive Learning**. To further inject the global multi-relational context into our recommender, we propose graph-level contrastive learning with heterogeneous hypergraphs. Specifically, in our component of parameterized heterogeneous hypergraph, we generate the hyperedge embedding by aggregating information from different users with the learnable user-intent hyperedge dependencies. Hence, the behavior-aware hyperedge representation can serve as graph readout information with the intent disentanglement. In real-life recommendation scenarios, users may share similar intents (*e.g.*, cares more about product price or brand) to interact with items in different ways (*e.g.*, page view, purchase, favorite).

In light of the intent commonality over the behavior heterogeneity, a graph-level contrastive regularization is designed for DHCF to capture the global multi-behavior context. In particular, we first aggregate the disentangled intent-aware information over the set of all type-specific hyperedge embeddings with sum-pooling, to generate the behavior-specific hyperedge representation $\bar{\boldsymbol{\Gamma}}_k^{(u)}$ at the global graph-level. Then, we treat the target-auxiliary behavior pairs as the positive contrasting instances. Motivated by the negative sample generation strategy in graph infomax (Velickovic et al., 2019; Jing et al., 2021), the graph-level negative pairs are constructed by randomly shuffling the learned hypergraph adjacent matrix to output the corrupted user embeddings $\bar{\boldsymbol{\Gamma}}_k^{'(u)}$. Formally, the graph-level multi-relational contrastive regularization is introduced with the following loss:

$$\mathcal{L}_g = -\sum_k \log \frac{\exp(s(\bar{\boldsymbol{\Gamma}}_{k'}, \bar{\boldsymbol{\Gamma}}_k))}{\exp(s(\bar{\boldsymbol{\Gamma}}_{k'}, \bar{\boldsymbol{\Gamma}}_k)) + \exp(s(\bar{\boldsymbol{\Gamma}}_{k'}', \bar{\boldsymbol{\Gamma}}_k))} \tag{8}$$

With the designed multi-relation contrastive learning, our DHCF method performs adaptive contrastive SSL to augment multi-behavior dependency modeling from sparse heterogeneous interactions, which brings benefits for enhancing the model robustness and recommendation performance.

## 3.7 MODEL OPTIMIZATION

With the prediction score $\hat{\mathcal{X}}_{i,j,k} = \mathbf{\Psi}_i^{(u)\top} \cdot \mathbf{\Psi}_j^{(v)\top}$, the pair-wise marginal loss function is applied into the objective optimization. Specifically, we randomly sample $S$ positive and negative sample pairs for each user, from his/her interacted and non-interacted items under the target behavior type. The overall optimized objective, which integrates the recommendation loss and the node-level and graph-level contrastive SSL objectives is formally given as follows:

$$\mathcal{L} = \sum_{i=1}^{N} \sum_{s=1}^{S} \max(0, 1 - (\hat{\mathcal{X}}_{i,p_s,k'} - \hat{\mathcal{X}}_{i,n_s,k'})) + \lambda_1 \cdot \|\mathbf{\Theta}\|_F^2 + \lambda_2 \cdot \mathcal{L}_n + \lambda_3 \cdot \mathcal{L}_g \qquad (9)$$

where $\lambda_1, \lambda_2, \lambda_3$ are weights to respectively control the influences of weight-decay regularization, node-level and graph-level contrastive regularizations during the model training. Algorithm 1 summarizes the learning process of our proposed DHCF in the Supplementary Section.

**Model Complexity Analysis**. We analyze DHCF's time complexity from three key modules: i) The complexity of multiplex graph relation learning is $O(|\mathcal{X}| \times d \times L)$, where $|\mathcal{X}|$ denotes the number of non-zero elements in $\mathcal{X}$. ii) The parameterized multi-behavior hypergraph neural network takes $O((I + J) \times E \times d)$ for message passing. Here, $E$ represents the number of latent factors for intent disentanglement. iii) For the component of contrastive SSL, the contrastive loss is calculated at the batch level, which takes $O(B^{(u)} \times B^{(v)} \times d)$ complexity, in which $B^{(u)}$ and $B^{(v)}$ denotes the number of users and items in a single batch, respectively. Based on the above analysis, our DHCF can achieve competitive time complexity with some state-of-the-art GNN-based recommenders.

## 3.8 IN-DEPTH DISCUSSION OF DHCF

**i) Adaptive Self-Supervision of DHCF**. In this part we show that, in comparison to vanilla GNNs, our heterogeneous hypergraph message passing mechanism can not only generate more supervision signals for the underlying id-corresponding embeddings, but also provide learnable weights to enable adaptive CF training. **ii) Rationale of Graph Multi-Relational CL**. We analyze the training objective for our graph-level contrastive learning (*i.e.* Eq 8), to show that it adaptively and efficiently maximizes the cross-relation similarity between nodes according to their global connectivity (*i.e.* how strong the nodes are connected to the global hyperedges). Our shuffling-based negative sampling essentially conducts uniform similarity minimization as the contrast of the positive optimization. The theoretical analysis with derivation details for the above two points are provided in Appendix A.5.

## 4 EVALUATION

### 4.1 EXPERIMENTAL SETUP

**Dataset**. Our experiments utilize three datasets, *i.e.*, Beibei, Tmall and IJCAI. **Beibei**. This dataset is collected from one popular e-commerce platform for maternal and infant product. The number of contained page view, add-to-cart, purchase interactions are 2,412,586; 642,622; 282,860 respectively. **Tmall**. It is a benchmark dataset for multi-behavior recommendation with four types of user-item interaction behaviors, *i.e.*, page view (4,542,043), add-to-favourite (201,402), add-to-cart (516,117), purchase (491,870). **IJCAI**. This large-scale data is released by IJCAI competition for user online activity modeling from an online retailing site. It shares the same set of behavior types with Tmall dataset, *i.e.*, page view (30,287,317), add-to-favourite (2,934,022), add-to-cart (74,168), purchase (2,926,616). More details on the used datasets can be found in Appendix A.1.

**Baseline Methods**. To conduct comprehensive evaluations on the effectiveness of DHCF, six groups of 18 baselines are included for performance comparison. **1)** Conventional CF Methods: **MF** (Koren et al., 2009) and **NCF** (He et al., 2017). **2)** Autoencoder/Autoregressive CF: **CDAE** (Wu et al., 2016) and **NADE** (Zheng et al., 2016). **3)** GNN-enhanced CF Models: **NGCF** (Wang et al., 2019) and **SGCN** (Zhang et al., 2019). **4)** Recommendation with Disentangled Representations: **DGCF** (Wang et al., 2020a), **GDCF** (Zhang et al., 2022), and **ICL** (Chen et al., 2022). **5)** Heterogeneous Recommendation: **NMTR** (Gao et al., 2019), **EHCF** (Chen et al., 2020a), **MGCN** (Jin et al., 2020), **GNMR** (Xia et al., 2021a), **KHGT** (Xia et al., 2021b), and **SMRec** (Gu et al., 2022). **6)** SSL for Recommendation: **SGL** (Wu et al., 2021a), **MHCN** (Yu et al., 2021), and **HCCF** (Xia et al., 2022). For a detailed description of all the baselines used in our experiments, please refer to Appendix A.4.

Table 1: Performance comparison over different datasets in terms of *HR@10* and *NDCG@10*.

| Data | Metric | MF | NCF | CDAE | NADE | NGCF | SGCN | DGCF | GDCF | ICL | *DHCF* |
|------|--------|------|------|------|------|------|------|------|------|------|--------|
| Beibei | HR | 0.588 | 0.594 | 0.608 | 0.608 | 0.611 | 0.609 | 0.612 | 0.623 | 0.645 | **0.679** |
| | NDCG | 0.333 | 0.338 | 0.341 | 0.343 | 0.369 | 0.343 | 0.344 | 0.377 | 0.396 | **0.419** |
| Tmall | HR | 0.265 | 0.306 | 0.326 | 0.332 | 0.321 | 0.339 | 0.395 | 0.363 | 0.421 | **0.525** |
| | NDCG | 0.165 | 0.174 | 0.193 | 0.194 | 0.191 | 0.191 | 0.267 | 0.211 | 0.278 | **0.325** |
| IJCAI | HR | 0.285 | 0.459 | 0.455 | 0.469 | 0.461 | 0.452 | 0.478 | 0.499 | 0.517 | **0.611** |
| | NDCG | 0.185 | 0.294 | 0.288 | 0.304 | 0.292 | 0.285 | 0.306 | 0.331 | 0.358 | **0.418** |

| Data | Metric | NMTR | EHCF | MGCN | GNMR | KHGT | SMRec | MHCN | SGL | HCCF | *DHCF* |
|------|--------|------|------|------|------|------|-------|------|------|------|--------|
| Beibei | HR | 0.613 | 0.633 | 0.642 | 0.623 | 0.640 | 0.610 | 0.614 | 0.619 | 0.610 | **0.679** |
| | NDCG | 0.349 | 0.384 | 0.376 | 0.358 | 0.385 | 0.354 | 0.345 | 0.346 | 0.354 | **0.419** |
| Tmall | HR | 0.361 | 0.370 | 0.478 | 0.461 | 0.468 | 0.359 | 0.411 | 0.413 | 0.374 | **0.525** |
| | NDCG | 0.206 | 0.210 | 0.273 | 0.261 | 0.282 | 0.207 | 0.249 | 0.261 | 0.215 | **0.325** |
| IJCAI | HR | 0.481 | 0.556 | 0.463 | 0.541 | 0.552 | 0.482 | 0.504 | 0.484 | 0.487 | **0.611** |
| | NDCG | 0.304 | 0.408 | 0.277 | 0.338 | 0.359 | 0.305 | 0.332 | 0.316 | 0.317 | **0.418** |

**Evaluation Protocols and Hyperparameter Settings**. We adopt two representative metrics for evaluating the accuracy of top-$N$ item recommendations: Hit Ratio ($HR@N$) and Normalized Discounted Cumulative Gain ($NDCG@N$). Following similar settings in Xia et al. (2021b); Jin et al. (2020), the leave-one-out evaluation strategy is applied to construct the test set with the users' last interactions under the target behavior type. The model is implemented with Tensorflow and optimized using Adam. The number of latent intent hyperedges for each behavior type is selected from {32,64,128,256,512}. During the training phase, the batch size and dropout ratio are chosen from {32, 64, 128, 256, 512} and {0.2, 0.4, 0.6, 0.8}, respectively. The number of message passing layers for all graph-based methods is tuned from {1,2,3}. The parameter $\lambda_1$ is tuned from the value range {$1e^{-5}$, $1e^{-4}$, $1e^{-3}$, $1e-2$}. The regularization strength of node-level and graph-level contrastive objectives is determined by $\lambda_2$ and $\lambda_3$ which are tuned from {$1e^{-7}$, $1e^{-6}$, $1e^{-5}$, $1e^{-4}$}.

## 4.2 PERFORMANCE COMPARISON

Based on the results in Table 1 and Table 4 (Appendix A.2), we draw the following observations:

- Our proposed DHCF consistently outperforms all groups of baselines by a significant margin, with a maximum p value of $1.6e^{-5}$. The performance of multi-behavior methods is better than that of single-behavior approaches, which justifies the advantage of incorporating heterogeneous behavior characteristics for user preference representation. The performance improvement between our DHCF and the state-of-the-art multi-behavior recommender systems verifies the benefits of intent disentanglement representations under the heterogeneous CF scenario. Additionally, our designed multi-behavior contrastive hypergraph module enables the effective modeling of global heterogeneous collaborative relations, which further boosts the performance.

- Our proposed DHCF outperforms the disentangled recommendation models, demonstrating that learning latent intent factors behind user preferences with only a single type of interaction behavior may be insufficient to distill the heterogeneous behavior characteristics. By incorporating behavior heterogeneity and diversity into the intent disentanglement, our DHCF significantly surpasses DGCF and GDCF in all cases through the designed hierarchical contrastive learning component over the heterogeneous hypergraph. This superior performance underscores the importance of considering the diverse user behavior patterns in building effective and accurate recommenders.

- While some recent studies (*e.g.*, SGL, HCCF) propose to perform augmentation to alleviate the sparse interaction issue, these works are specifically designed for homogeneous user-item relationships, due to the computational simplicity. To identify the latent factors behind sparse multi-behavior user-item interactions, we design a heterogeneous hypergraph contrastive learning paradigm with multi-channel intent representation space for adaptive data augmentation, to realize the disentangled representation with behavior heterogeneity and diversity.

## 4.3 ABLATION STUDY OF DHCF

To investigate the benefits of key components in DHCF, we perform an ablation study with variants:

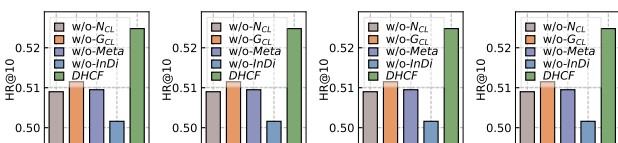

Figure 4: Hyperparameter study of the DHCF model.

- **w/o-$N_{CL}$**: We exclude the component with the designed node-level adaptive contrastive learning, which provides self-supervised signals with the user's multi-behavior self-discrimination.
- **w/o-$G_{CL}$**: We remove the graph-level multi-behavior contrastive learning from DHCF for capturing global collaborative context of multiplex user-item relationships.
- **w/o-$Meta$**: We disable the adaptive contrastive projection to encode the personalized multi-behavior semantics of different users based on the designed meta network.
- **w/o-$InDi$**: This variant does not enable the intent disentanglement with multi-channel parameterized hypergraphs to generate disentangled representations on heterogeneous behavior data.

The results of our ablation study are shown in Figure 2. We observe that our DHCF achieves the best performance, demonstrating that all key components contribute to learning expressive representations. Compared with w/o-$N_{CL}$ and w/o-$G_{CL}$,

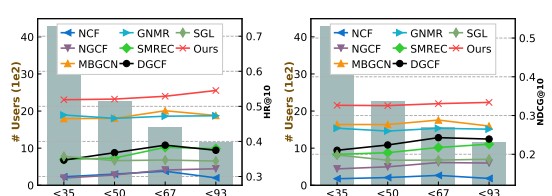

Figure 2: Model ablation study on Tmall and IJCAI.

we find that performing hierarchical contrastive learning over the multi-relational interaction data indeed helps the model to encode better user embeddings with both local and global heterogeneous collaborative context. Additionally, our DHCF can perform contrastive SSL in an adaptive manner to encode the customized behavior semantics for self-supervised augmentation, which benefits performance compared to w/o-$Meta$. The performance of w/o-$InDi$ is improved by our DHCF, verifying the effectiveness of user multi-behavior representation learning in multiple intents.

## 4.4 MODEL ROBUSTNESS ANALYSIS

We investigate the robustness of our DHCF against interaction data sparsity in recommender systems. We generate four user groups in terms of different user interaction frequency (*e.g.*, [35,50], [50,67)) and perform the evaluation separately. Compared with several representative baselines, our DHCF consistently achieves the best recommendation accuracy under different

Figure 3: Performance *w.r.t* user interaction frequency.

interaction frequencies, which confirms that our heterogeneous hypergraph self-augmentation at both node and graph levels, is beneficial for improving recommendation when the user interaction labels are limited. While MBGCN and GNMR attempt to leverage multiple types of user-item interactions to address the data scarcity issue, they still suffer from the sparse supervision label of limited target behavior data, leading to suboptimal representations over sparse user behaviors and long-tail items.

## 4.5 HYPERPARAMETER SENSITIVITY

We evaluate the parameter influence, including the hidden state dimensionality $d$, the number of latent intent hyperedges $E$, and the number of message passing iterations over both the multi-behavior graph and hypergraph structures. The results, presented in Figure 4, show that: (1) The larger hidden state dimensionality size may result in overfitting on sparse user and item representations. (2) With an increase in the number of intent representation channels, our DHCF's performance first improves and then deteriorates. This observation suggests that redundant intent hyperedges may introduce noise and hinder disentangled representation. (3) Our DHCF can benefit from stacking two message passing layers. Generally, with the help of hypergraph-enhanced global relation modeling, two embedding propagation layers are sufficient to capture the high-order connectivity of heterogeneous interaction

Table 2: Efficiency comparison in terms of computational time (seconds).

| Data | DGCF | HCCF | KHGT | GNMR | SMRec | DHCF | DGCF | HCCF | KHGT | GNMR | SMRec | DHCF |
|---|---|---|---|---|---|---|---|---|---|---|---|---|
| | Model Training Time (s) | | | | | | Model Testing Time (s) | | | | | |
| Beibei | 11.2 | 15.3 | 22.3 | 9.6 | 36.1 | **5.8** | 9.2 | 7.2 | 23.2 | 6.1 | 38.4 | **4.5** |
| Tmall | 37.7 | 29.0 | 41.1 | 25.6 | 51.1 | **17.2** | 30.1 | 32.2 | 36.0 | 24.0 | 48.7 | **21.3** |
| IJCAI | 43.5 | 35.1 | 49.3 | 38.3 | 59.6 | **30.4** | 39.9 | 41.2 | 42.3 | 34.5 | 54.5 | **37.9** |

signals. Further increasing the depth of DHCF from three to four might lead to the embedding over-smoothing issue, which is harmful for differentiating diverse user preferences.

## 4.6 MODEL EFFICIENCY STUDY

To validate the efficiency of our DHCF method, we compare DHCF against several state-of-the-art baseline methods for evaluating their computational cost, in terms of model training and testing time of each epoch on different datasets. From the results presented in Table 2, DHCF consistently performs better than compared baselines in efficiency, with less training and inference time. Compared with the best performed (measured by recommendation accuracy) baseline KHGT, our parameterized hypergraph encoder not only learns more informative user preference with intent-aware multi-behavior patterns, but also improves the model efficiency with a lightweight heterogeneous GNN architecture. While DGCF and HCCF focuses on modeling single type of user-item interactions, better efficiency can be achieved in our DHCF with interaction heterogeneity. Such advantages should be ascribed to the efficient architecture of our hypergraph neural network and graph-level contrastive learning.

## 4.7 CASE STUDY

To show the benefits of our DHCF model in capturing the global user dependencies and personalized multi-behavior relationships, we provide some cases shown in Figure 5. We project the encoded user embeddings into different colors, in which highly correlated embeddings are represented with similar colors. While user pairs $(u_{15}, u_{34})$ and $(u_{23}, u_{27})$ are not directly connected on the multiplex interaction graph $\mathcal{G}$, we can observe that the learned embeddings of exhibit strong dependencies, due to

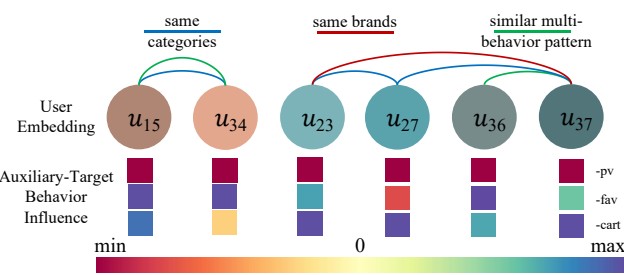

Figure 5: Case study of capturing global user dependencies and multi-behavior relationships in latent embedding space.

the same categories or brands of their interacted products (from external item side knowledge). This observation justifies the capability of our DHCF in capturing the global collaborative relations among users with the exploration of intent disentanglement and behavior heterogeneity. In addition, we visualize the relevance weights between the auxiliary (*e.g.*, page view) and target behaviors (*i.e.*, purchase) learned from our adaptive meta network. From the visualization results, our DHCF is capable of encoding the latent user dependencies $(u_{15}, u_{34})$ and $(u_{23}, u_{27})$, *w.r.t* their similar and interpretable interaction influence between the auxiliary and target behaviors.

## 5 CONCLUSION

In this paper, we address the new problem of disentangled heterogeneous collaborative filtering. Specifically, Our proposed DHCF focuses on learning heterogeneous factorized representations to effectively disentangle latent intents across diverse user interactions. To enhance the expressiveness and robustness of the encoded embeddings, we introduce a hierarchical contrastive learning approach. This approach enables our DHCF to perform adaptive augmentation over parameterized heterogeneous hypergraph structures. We conduct experiments on publicly available datasets to demonstrate the superiority of our DHCF approach. Our future work aims to explore the potential of pre-training strategies with our model for representation enhancement across diverse recommender systems.

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

## A   APPENDIX

### A.1   THE DETAILS OF DATASETS

**Beibei** is a dataset about maternity and baby products on the Chinese e-commerce platform[1], which contains sales data from January 2017 to April 2021. The dataset includes more than 500,000 products, covering various categories such as formula milk, diaper rash cream, baby clothes, and so on. It provides detailed information about the products and user infomation, such as product brand, price and sales volume, and user ID, user reviews and purchase time. In addtion, it also contains three behavior types(such as *page views, favorites and purchases*).

**Tmall** is a dataset provided by Taobao, one of the largest B2C e-commerce platforms in China, which covers the daily sales and user behavior from November 2014 to December 2014. The dataset contains data from over 120 million users and millions of transactions, including product information, user behavior, and user personal information. Additionally, the dataset provides multiple interaction behavior records(*e.g., page views, favorites, add-to-carts, and purchases*).

**IJCAI** was released by the IJCAI Contest[2] 2015, and used for the task of predicting repeat buyers. The dataset contains 6 months of anonymous shopping logs of users before and after the Double 11 event, as well as label information indicating whether the user is a repeat buyer. The dataset contains four behavior types( *clicks, favorites, add-to-carts, and purchases*).

The summary table characterizes those three datasets as following:

Table 3: Statistics of the Experimented Datasets

| Dataset | User # | Item # | Interaction # | Sparsity# | behavior types# |
|---------|--------|--------|---------------|-----------|-----------------|
| Beibei | 21,716 | 7,977 | 3,338,068 | 0.9807 | page views, favorites and purchases |
| Tmall | 114,503 | 66,706 | 5,751,432 | 0.9992 | page view, favorites, add-to-carts, and purchases |
| IJCAI | 423,423 | 874,328 | 36,222,123 | 0.9999 | clicks, favorites, add-to-carts, and purchases |

### A.2   PERFORMANCE COMPARISON ON TOP-$N$ ITEM POSITIONS

To fully verify the effectiveness of the experiment, we also conducted experiments on the Top-N recommendation task for different values of $N$. Table 4 presents the evaluation results on the Beibei dataset in terms of NDCG@N. From the results, it can be observed that when $N$ takes different values from the set {1,3,5,7,9}, DHCF consistently achieves the best performance, further verifying the effectiveness of the proposed heterogeneous hypergraph neural network multi-behavior recommendation model combined with multi-behavior contrastive learning.

Table 4: The best performed baselines from each category on Beibei are reported in this table.

| Model | NDCG@1 | NDCG@3 | NDCG@5 | NDCG@7 | NDCG@9 |
|-------|--------|--------|--------|--------|--------|
| MF | 0.1184 | 0.2275 | 0.2866 | 0.3164 | 0.3293 |
| NCF | 0.1228 | 0.2316 | 0.2834 | 0.3154 | 0.3300 |
| ICL | 0.1636 | 0.3170 | 0.3552 | 0.3787 | 0.3887 |
| SGL | 0.1252 | 0.2357 | 0.2962 | 0.3288 | 0.3381 |
| EHCF | 0.1775 | 0.3029 | 0.349 | 0.3724 | 0.3887 |
| GNMR | 0.1395 | 0.2567 | 0.3086 | 0.3334 | 0.3515 |
| **DHCF** | **0.1967** | **0.3224** | **0.3748** | **0.3992** | **0.4142** |

### A.3   THE DETAILS OF MODEL OPTIMIZATION

We elaborate the overall workflow of training our DHCF framework in Algorithm 1 as follows. This algorithm summarizes and generalizes the learning process of the DHCF proposed, and provides a brief description of the entire recommendation task optimization.

---

[1]https://www.beibei.com/

[2]https://tianchi.aliyun.com/dataset/

---

**Algorithm 1:** The Learning Process of DHCF Framework

---

**Input:** User-item heterogeneous interactions $\mathcal{X} \in \mathbb{R}^{I \times J \times K}$, target behavior $t$, auxiliary behavior $a$, maximum epoch number $S$, number of graph iterations $L$, learning rate $\eta$, regularization weight $\lambda_1$, $\lambda_2$, $\lambda_3$.

**Output:** Trained model parameters $\Theta$.

1 Initialize model parameters $\Theta$
2 **for** $s \leftarrow 1$ *to* $S$ **do**
3      Draw a mini-batch $\mathbf{U}$ from all users $\{1, 2, ..., I\}$
4      Sample $m$ positive items $\{v_{p_1}, ..., v_{p_m}\}$
5      Sample $m$ negative items $\{v_{n_1}, ..., v_{n_m}\}$
6      **for** *each* $u_i \in \mathbf{U}$ **do**
7          Initialize the training loss $\mathcal{L} = \lambda_1 \cdot \|\Theta\|_{\mathrm{F}}^2$
8          **for** $l \leftarrow 1$ *to* $L$ **do**
9              Conduct the type-aware message passing (Eq 1)
10              Compute $\Lambda^{(u,1)}$ according to Eq 2 to Eq 4
11          **end**
12          Aggregating multi-order representations $\Lambda^{(u,1)}$ from $L$ iterations as $\Psi^{(u)}$ (Eq 5)
13          Calculate the prediction score $\hat{\mathcal{X}}_{i,j,k} = \Psi_i^{(u)\top} \cdot \Psi_j^{(v)\top}$
14          $\mathcal{L}+ = \sum_{m=1}^{M} \max(0, 1 - (\hat{\mathcal{X}}_{i,p_s,k'} - \hat{\mathcal{X}}_{i,n_s,k'}))$
15          **for** $l \leftarrow 1$ *to* $L$ **do**
16              Calculate the node-level infoNCE loss $\mathcal{L}_n^l$ (Eq 7)
17              Calculate the graph-level constractive loss $\mathcal{L}_g^l$(Eq 8)
18          **end**
19          Calculate the recommendation loss, weight-decay regularization, the node and graph level loss to obtain the overall loss $\mathcal{L}$ (Eq 9)
20          **for** *each parameter* $\theta$ *in* $\Theta$ **do**
21              $\theta = \theta - \eta \cdot \frac{\partial \mathcal{L}}{\partial \theta}$
22          **end**
23      **end**
24 **end**
25 **return** all parameters $\Theta$

---

### A.4 THE DETAILED DESCRIPTION OF BASELINE METHODS

**1) Conventional Collaborative Filtering Models**.

- **MF** (Koren et al., 2009): This baseline is a matrix factorization approach which incorporates user and item bias information for the implicit feedback records between users and items.

- **NCF** (He et al., 2017): This is a milestone work on neural collaborative filtering, utilizing the deep multi-layer perceptron (MLP) to enable non-linear feature interaction extraction.

**2) Autoencoder/Autoregressive Collaborative Filtering**.

- **CDAE** (Wu et al., 2016): This framework enhances CF model with the reconstruction-based optimization loss using the denoising auto-encoder network with input data corruption.

- **NADE** (Zheng et al., 2016): It is an autoregressive method which shares the parameters among different user-item interactions. Multiple hidden layers are in used in NADE for transformation.

**3) GNN-enhanced Collaborative Filtering Methods**.

- **NGCF** (Wang et al., 2019): This representative graph neural network model captures the high-order relationships among users and items with recursive message passing for representation updating.

- **SGCN** (Zhang et al., 2019): This approach stacks multiple encoder-decoders over the GNN architecture with the embedding reconstruction loss to address the data sparsity issue. The reconstruction loss is applied over the encoded latent embeddings with value masking.

**4) Recommendation with Disentangled Representations**.

- **DGCF** (Wang et al., 2020a): This baseline method is a disentangled collaborative filtering approach for encoding latent factors over the user-item interaction graph, built upon the GCN propagation scheme. Each user representation is partitioned into intent-aware embedding vectors to represent latent factors driving users' interaction behaviors over items.

- **GDCF** (Zhang et al., 2022): This recent disentangled recommender in which multi-typed geometries are incorporated into interaction disentanglement for generating factorized embeddings.

- **ICL** (Chen et al., 2022): It proposes to improve the model robustness by leveraging contrastive self-supervised learning for modeling latent intent distributions over the item-interacted behaviors.

**5) Multi-Behavior Recommender Systems**.

- **NMTR** (Gao et al., 2019): It is a multi-task learning model which captures the correlations of different types of interactions in recommener system with pre-defined cascaded behavior relations.

- **EHCF** (Chen et al., 2020a): It tackles the heterogeneous collaborative filtering with a non-sampling transfer learning approach, to correlate behavior-aware predictions for recommendation.

- **MBGCN** (Jin et al., 2020): This recommendation model is built over the iterative graph message passing paradigm to propagate the behavior-aware embeddings over the heterogeneous interaction graph to model multi-typed user-item relations.

- **GNMR** (Xia et al., 2021a): The self-attention is integrated with a memory network to jointly encode the behavior-specific semantics and behavior-wise dependencies. Low-order user embeddings are selectively to be combined with high-order representations.

- **KHGT** (Xia et al., 2021b): This baseline is another multi-behavior recommendation approach, which utilizes the stacked graph transformer network to aggregate behavior-aware representations through attentive weights for differentiating propagated messages.

- **MRec** (Gu et al., 2022): This baseline method designs a star-style contrastive learning task to model the correlations between the target behavior and the auxiliary behaviors of users.

**6) Self-Supervised Recommendation Models**.

- **SGL** (Wu et al., 2021a): This method proposes to augment user-item interaction graph with random walk-based node and edge dropout operators for constructing contrastive views.

- **MHCN** (Yu et al., 2021): A generative self-supervised task is incorporated into recommendation loss by maximizing mutual information between path-level and global-level embeddings. It considers high-order user correlations with hypergraph convolutions.

- **HCCF** (Xia et al., 2022): It is a hypergraph contrastive learning model for generating self-supervised signals with local-global node self-discriminating. A parameterized hypergraph neural module is developed to aggregate information from user and item individuals with hyperedges.

## A.5 IN-DEPTH DISCUSSION ABOUT MULTI-RELATION ADAPTIVE CONTRASTIVE LEARNING

### A.5.1 Adaptive Self-Supervision of DHCF

In this section we show that, in comparison to vanilla GNNs, our heterogeneous hypergraph message passing mechanism can not only generate more supervision signals for the underlying id-corresponding embeddings, but also provide learnable weights to enable adaptive CF training. In specific, the key of the pair-wise recommendation loss (*i.e.* the first term in Eq 9) is to maximize or minimize the prediction scores $\hat{\mathcal{X}}_{i,j,k'}$ for positive or negative training pairs $(u_i, v_j)$, respectively. For vanilla GNNs, the prediction using the $L$-th order embeddings can be decomposed as follows:

$$\hat{\mathcal{X}}^{\mathrm{G}}_{i,j,k'} = \mathbf{z}_i^\top \mathbf{z}_j = \left( \sum_{i' \in \mathcal{N}_i^L} \alpha_{i'} \mathbf{e}_{i'} \right)^\top \cdot \left( \sum_{j' \in \mathcal{N}_j^L} \alpha_{j'} \mathbf{e}_{j'} \right) \tag{10}$$

$$= \sum_{i' \in \mathcal{N}_i^L} \sum_{j' \in \mathcal{N}_j^L} \alpha_{i'} \alpha_{j'} \cdot \mathbf{e}_{i'}^\top \mathbf{e}_{j'}$$

where $\hat{\mathcal{X}}^{\mathrm{G}}_{i,j,k'}$ denotes the prediction for $(u_i, v_j)$ under the target behavior $k'$, using a GNN model. $\mathbf{z}_i, \mathbf{z}_j \in \mathbb{R}^d$ denote the high-order embeddings for $u_i$ and $v_j$ given by the GNN. $\mathcal{N}^L_i, \mathcal{N}^L_j$ denote the set of neighboring nodes in $L$ hops for $u_i$ and $v_j$, respectively. $\mathbf{e}_i, \mathbf{e}_j \in \mathbb{R}^d$ represent the learnable parameters for the id-corresponding embeddings of $u_i$ and $v_j$, respectively. As shown by the above decomposition, when maximizing or minimizing $\hat{\mathcal{X}}^{\mathrm{G}}_{i,j,k'}$, it does not merely adjust the prediction for the concerned two nodes $u_i$ and $v_j$. The nodes in $L$ hops to $u_i$ and $v_j$ are all pulled closer or pushed away in their underlying embeddings. The strength of these optimization terms are determined by the coefficients $\alpha_{i'}, \alpha_{j'}$, which are related to the degrees of nodes in the heterogeneous interaction graph. Although the foregoing GNN-based embeddings implicitly augment the supervision signals for the neighboring node pairs, we show that by employing our heterogeneous hypergraph architecture for representation learning, even more supervision signals can be generated, also with learnable strength coefficients, to further enhance the parameter learning for better graph relation modeling. Analogous to Eq 10, the prediction score $\hat{\mathcal{X}}^{\mathrm{H}}_{i,j,k'}$ for $(u_i, v_j)$ under the target behavior $k'$ made by our heterogeneous hypergraph neural networks can be decomposed as follows:

$$
\begin{aligned}
\hat{\mathcal{X}}^{\mathrm{H}}_{i,j,k'} = \mathbf{H}^\top_i \mathbf{H}_j &= \left( \sum_{e=1}^{E} \delta \left( \tilde{\mathcal{H}}_{i,e} \cdot \sum_{i'=1}^{I} \delta \left( \tilde{\mathcal{H}}_{i',e} \cdot \mathbf{e}_{i'} \right) \right) \right)^\top \\
&\cdot \left( \sum_{e=1}^{E} \delta \left( \tilde{\mathcal{H}}_{j,e} \cdot \sum_{j'=1}^{J} \delta \left( \tilde{\mathcal{H}}_{j',e} \cdot \mathbf{e}_{j'} \right) \right) \right) \\
&= \sum_{i'=1}^{I} \sum_{j'=1}^{J} \beta_{i'} \beta_{j'} \cdot \mathbf{e}^\top_{i'} \mathbf{e}_{j'}
\end{aligned}
\tag{11}
$$

where $\hat{\mathcal{X}}^{\mathrm{H}}_{i,j,k'}$ denotes the prediction for $(u_i, v_j)$ under the target behavior $k'$ with our heterogeneous hypergraph architecture. $\mathbf{H}_i, \mathbf{H}_j \in \mathbb{R}^d$ denote the high-order embeddings for $u_i$ and $v_j$ through the hypergraph neural network (HGNN). For simplicity, we assume $\delta(\cdot)$ is the identity function. $\beta_{i'}, \beta_{j'}$ denote the learnable HGNN-based weights for simplifying the equations. As shown by the above decomposition, our hypergraph networks not only adjust the predictions for node pairs in the $L$-hop neighborhood, but also generates supervision signals for nodes from the global graph level, which is far more in amount than vanilla GNNs. This shows that our hypergraph relation learning is able to conduct graph-level supervision signal enrichment. Furthermore, the weights $\beta_{i'}, \beta_{j'}$ for each term are calculated and optimized by the hypergraph neural networks, which supercharge our DHCF framework with more capability of adaptive relation learning.

### A.5.2 Rationale of Graph Multi-Relational CL

In this section, we analyze the training objective for our graph-level contrastive learning (*i.e.* Eq 8), to show that this training objective adaptively and efficiently maximizes the cross-relation similarity between nodes according to their global connectivity (*i.e.* how strong the nodes are connected to the global hyperedges). Our shuffling-based negative sampling essentially conducts uniform similarity minimization as the contrast of the positive optimization. Without loss of generality, we simplify Eq 8 by using dot-product as the similarity function $s(\cdot)$, to have the following loss:

$$
\mathcal{L}_g = \sum_{k=1}^{K} -\bar{\mathbf{\Gamma}}^\top_{k'} \bar{\mathbf{\Gamma}}_k + \log \left( \exp(\bar{\mathbf{\Gamma}}^\top_{k'} \bar{\mathbf{\Gamma}}_k) + \exp(\bar{\mathbf{\Gamma}}'^\top_{k'} \bar{\mathbf{\Gamma}}_k) \right)
\tag{12}
$$

where $-\bar{\mathbf{\Gamma}}^\top_{k'} \bar{\mathbf{\Gamma}}_k$ represents the positive term that pulls close the averaged embeddings of hyperedges in the target behavior $k'$ and the averaged hyperedge embedding in the auxiliary behavior types $k$. And $\bar{\mathbf{\Gamma}}'^\top_{k'} \bar{\mathbf{\Gamma}}_k$ denotes the negative term that pushes away the averaged hyperedge embedding $\bar{\mathbf{\Gamma}}'_{k'}$ of the negative sample and the vanilla averaged hyperedge embedding. Here the negative sample refers to the randomly-shuffled graph. Then we individually analyze the positive term and the negative term,

by decomposing them into low-level node embeddings as shown below:

$$-\bar{\mathbf{\Gamma}}_{k'}^\top \bar{\mathbf{\Gamma}}_k = -\left(\sum_{e_1=1}^{E}\sum_{i_1=1}^{I} \tilde{\mathcal{H}}_{i_1,e_1,k'} \cdot \mathbf{e}_{i_1}\right)^\top \left(\sum_{e_2=1}^{E}\sum_{i_2=1}^{I} \tilde{\mathcal{H}}_{i_2,e_2,k} \cdot \mathbf{e}_{i_2}\right)$$

$$= -\sum_{e_1,e_2}\sum_{i_1,i_2} \tilde{\mathcal{H}}_{i_1,e_1,k'} \tilde{\mathcal{H}}_{i_2,e_2,k} \cdot \mathbf{e}_{i_1}^\top \mathbf{e}_{i_2}$$

$$\bar{\mathbf{\Gamma}}_{k'}^{'\top} \bar{\mathbf{\Gamma}}_k = \left(\sum_{e_1=1}^{E}\sum_{i_1=1}^{I} \epsilon \cdot \mathbf{e}_{i_1}\right)^\top \left(\sum_{e_2=1}^{E}\sum_{i_2=1}^{I} \tilde{\mathcal{H}}_{i_2,e_2,k} \cdot \mathbf{e}_{i_2}\right)$$

$$= \sum_{e_1,e_2}\sum_{i_1,i_2} \epsilon \cdot \tilde{\mathcal{H}}_{i_2,e_2,k} \cdot \mathbf{e}_{i_1}^\top \mathbf{e}_{i_2}$$

For positive samples, the corresponding term $\partial - \bar{\mathbf{\Gamma}}_{k'}^\top \bar{\mathbf{\Gamma}}_k / \partial \mathbf{e}_{i_1}^\top \mathbf{e}_{i_2} = \tilde{\mathcal{H}}_{i_1,e_1,k'} \tilde{\mathcal{H}}_{i_2,e_2,k}$ indicates that our graph-level CL essentially maximizes the similarity between each node pair $(i_1, i_2)$, according to their individual relation (*i.e.* $\tilde{\mathcal{H}}$) to all the hyperedges. And $\partial - \bar{\mathbf{\Gamma}}_{k'}^\top \bar{\mathbf{\Gamma}}_k / \partial \tilde{\mathcal{H}}_{i_1,e_1,k'} \tilde{\mathcal{H}}_{i_2,e_2,k} = \mathbf{e}_{i_1}^\top \mathbf{e}_{i_2}$ shows that our graph-level CL also adaptively maximizes the cross-relation hypergraph structures for target relation $k'$ and auxiliary relation $k$ according to the similarity of node embeddings. For negative samples, one of the hypergraph weight is substituted by a noise coefficient $\epsilon$, which leads to similarity minimization with random strength. As there is no straightforward negative samples for graph-level hyperedge embeddings, this strategy enables generally uniform contrastive optimization against the similarity maximization for the positive samples.

