# OpenReview forum: "Disentangled Heterogeneous Collaborative Filtering"
_ICLR.cc/2024/Conference — Submitted to ICLR 2024_

### Official Review · Reviewer_KKy5 · 2023-10-27

**Soundness:** 2 fair
**Presentation:** 3 good
**Contribution:** 2 fair
**Rating:** 5
**Confidence:** 2

**Summary:**

In this work, the authors focus on combining intent disentanglement and multi-behavior modeling for collaborative filtering. The proposed method -- DHCF utilizes parameterized heterogeneous hypergraph to encode intents embeddings, and introduces behavior-wise contrastive learning to improve model robustness. Offline experiments are conducted on public datasets to demonstrate the performance of DHCF on Top-k item recommendation.

**Strengths:**

1. This paper provides insight on utilizing multi-behavior data in recommendation systems.
2. The proposed method outperforms the baseline methods on HR and NDCG for top-10 recommendation.
3. Ablation analysis is included.

**Weaknesses:**

1. The model consists of multiple components, which are difficult to optimize and converge. It is hard to be applied in the real-world case.
2. The robustness analysis is not convincing. To demonstrate the model's robustness, we expect it to achieve stable performance on difficult tasks, where baseline methods perform poorly compared to easy tasks. However, in Figure 3, the baseline methods achieve similar performance on different user groups. This comparison is not convincing evidence of the method's robustness.

**Questions:**

1. It is strange that the basic methods like NCF achieve similar performance on different user groups (in Figure 3). Instead of evaluating on tailed items, why did the authors test the performance on different user groups for robustness analysis?
2. Is the meta-learning process considered while calculating the complexity in section 3.7?
3. Any evidence to support the convergence of the learnable hypergraphs?

---

> ### Author Response · Authors · 2023-11-20
>
> Responses to Reviewer KKy5
>
> Thank you for your dedicated efforts and valuable feedback. We would like to address your concerns as follows:
>
> 1. The practicality of the proposed DHCF model (W1)
>
> - Model Efficiency. In our efficiency study, as presented in Section 4.6 and Table 2, we demonstrate that the per-epoch training and testing time of DHCF is comparable to existing baseline models. This indicates that DHCF is applicable in real-world scenarios and is efficient in terms of computational resources.
>
> - Model Convergence. Regarding the convergence of the model, our DHCF follows the same maximum training epochs as other baseline methods, which is set to 100 epochs. This suggests that there are no disadvantages in terms of convergence speed for DHCF compared to other models.
>
> 2. Further clarification of the NCF's results on different user groups (in Figure 3).  (W2 & Q1)
>
>     Thank you for your insightful feedback. We appreciate your comment and would like to address it with the following points:
>  - **Increment of supervision signals does not always improve performance**. The observation that basic methods like NCF achieve similar performance on different user groups, as shown in Figure 3, is consistent with findings from previous works on multi-behavior recommendation, such as [1-2]. This phenomenon highlights that simply increasing supervision signals does not necessarily lead to improved model performance. The lack of high-quality self-supervision signals can limit the effectiveness of increasing supervision, and this issue can also affect self-supervised learning methods based on random augmentations.
>  - **Consistent performance gain of DHCF**. In contrast to existing supervision-enhancing methods like SGL, our DHCF achieves performance improvement across different data sparsity levels through intent disentanglement and modeling of heterogeneous relation dependencies. This design effectively enhances the overall model performance. The consistent performance superiority of DHCF over the baseline models further validates the effectiveness of these designs.
>
>     [1] "Knowledge Enhancement for Contrastive Multi-Behavior Recommendation"
>
>     [2] "Graph Meta Network for Multi-Behavior Recommendation"
>
> 3. Performance evaluation on tailed items (Q1)
>
> - Thank you for your suggestion. We appreciate your interest in evaluating our model's performance on tailed items. In our experiment, we specifically considered a user set composed of long-tailed users to address the issue of sparsity. These users have a maximum of 35 interactions across different types, with an average of only 3.81 purchase interactions. This indicates that these users have a significantly low number of interactions. While we did not explicitly mention "tailed items" in our paper, our analysis does encompass the evaluation of our model's performance on this particular long-tailed user set.
>
> 4. Is the meta-learning process considered while calculating the complexity in section 3.7? (Q2)
> - We appreciate your thoughtful question. Although we did not explicitly mention the complexity of the meta network in our paper for the sake of simplicity, it involves a straightforward linear transformation. This transformation incurs a cost of $O((I+J)\times d^2)$ for each of the $K$ heterogeneous relations. Empirically, we have found that this complexity is close to $O(|\mathcal{X}|\times d)$, which is smaller than the complexity of the graph relation learning process. The graph relation learning process has a cost of $O(|\mathcal{X}|\times d\times L)$, where $|\mathcal{X}|$ represents the size of the dataset and $L$ denotes the number of graph convolutional layers.
>
> 5. Discussion of model convergence with the learnable hypergraphs? (Q3)
> - Thank you for your question. During the training process of DHCF, we have noticed that the performance on the validation set remains stable in the last few epochs. This consistent behavior can be replicated using the code we have released, providing strong evidence for the convergence of the learnable hypergraphs.

---

> > ### Comment · Reviewer_KKy5 · 2023-11-22
> > **Thank you for the responses**
> >
> > After carefully reading the responses and the comments from other reviewers, I would keep my score at this point.

---

### Official Review · Reviewer_7WGw · 2023-10-31

**Soundness:** 2 fair
**Presentation:** 3 good
**Contribution:** 2 fair
**Rating:** 3
**Confidence:** 2

**Summary:**

The paper addresses the problem of modern recommender systems that often utilize low-dimensional latent representations to embed users and items based on their observed interactions. However, many existing recommendation models are primarily designed for coarse-grained and homogeneous interactions, which limits their effectiveness in two key dimensions: i) They fail to exploit the relational dependencies across different types of user behaviors, such as page views, add-to-favorites, and purchases; and ii) They fail to disentangle the latent intent factors behind each behavior, which leads to suboptimal recommendations.

The authors argue that these limitations can be addressed by a novel recommendation model called Disentangled Heterogeneous Collaborative Filtering (DHCF). DHCF effectively disentangles users' multi-behavior interaction patterns and the latent intent factors behind each behavior. The authors propose a parameterized heterogeneous hypergraph architecture that captures the complex and diverse interactions among users, items, and behaviors. They also introduce a novel contrastive learning paradigm that improves the model's robustness against data sparsity.

**Strengths:**

1. The authors' approach is based on a hypergraph structure that allows for the modeling of multiple types of interactions among users and items. The hypergraph structure is parameterized, which means that the model can learn the weights of the hyperedges that connect users and items based on their interactions.

2. Contrastive learning is a technique that learns representations by contrasting positive and negative examples. In the context of DHCF, the authors use contrastive learning to learn representations of users and items that are optimized for predicting the interactions between them. By using contrastive learning, the authors are able to learn more robust representations that are less sensitive to data sparsity.

3. The authors' experiments show that DHCF significantly outperforms various strong baselines on three public datasets, which further supports the effectiveness of their approach.

**Weaknesses:**

1. The proposed Dynamic Hypergraph Collaborative Filtering (DHCF) approach presents a unique take on recommendation systems; however, its distinctiveness and advancements over existing methodologies in the literature are not sufficiently highlighted. To strengthen the paper, the authors should conduct a more comprehensive comparison of DHCF with prevailing models, pinpointing exact areas of improvement and innovation. Integrating and discussing the influence of more contemporary trends in recommendation systems, such as applications of deep learning or graph neural networks, would further enrich the paper's relevance and depth.

2. The paper currently lacks clarity and detail regarding the algorithms and techniques underpinning the DHCF approach. To remedy this, a more explicit elucidation of the methodology is required. Additionally, incorporating visual aids or concrete examples could help in visualizing the hypergraph structure and elucidating the concept of behavior-wise contrastive learning, making the paper more accessible and informative.

3. A more thorough examination of DHCF would contribute to a balanced and comprehensive understanding of the approach. Specific areas such as the scalability of DHCF to larger datasets and its sensitivity to hyperparameter choices warrant detailed discussion.

**Questions:**

1. In the paper, you mention that many existing recommendation models fail to exploit the relational dependencies across different types of user behaviors. Could you provide more details on how DHCF addresses this limitation? How does the parameterized heterogeneous hypergraph architecture capture the complex and diverse interactions among users, items, and behaviors?

2. In the paper, you also mention that many existing recommendation models fail to disentangle the latent intent factors behind each behavior. Could you provide more details on how DHCF disentangles the latent intent factors behind each behavior? How does the behavior-wise contrastive learning paradigm facilitate adaptive data augmentation at both the node and graph levels?

---

> ### Author Response · Authors · 2023-11-20
>
> Responses to Reviewer 7WGw
>
> Thank you for your kind feedback. We appreciate your concerns, and we would like to address them as follows:
>
> 1. Further clarifications of difference between the new model over existing methods. (W1)
>
> - We appreciate your feedback. In our work, we address two important challenges that have been overlooked in the field of collaborative filtering: interaction heterogeneity and fine-grained modeling of user intents. To bridge these gaps, we propose DHCF, a method that integrates intent disentanglement and multi-behavior modeling using a parameterized heterogeneous hypergraph architecture. Furthermore, we introduce a novel approach to heterogeneous contrastive learning, leveraging the encoded disentangled heterogeneous interaction patterns.
>
> 2. Further discussion of some details in the methodology part. (W2)
>
> - Thank you for your feedback. To better address your concern, we kindly request specific feedback on the parts that require further clarification. This will enable us to identify and improve those specific areas more effectively. Please let us know if there are any specific questions or sections where you would like us to provide more details.
>
> 3. Clarifications about the already conducted scalability and hyperparameter studies. (W3)
>
> - We appreciate your thoughtful suggestion. In our work, we have thoroughly investigated the influence of hyperparameter settings in Section 4.5 and examined model scalability in Section 4.6. Additionally, we have conducted comprehensive experiments covering various aspects. Specifically, we have performed a rigorous performance comparison with 18 diverse baselines, explored the impact of hyperparameter settings, conducted module ablation studies, assessed model scalability, evaluated robustness against data sparsity, and presented a detailed case study on the learned hypergraph structures.
>
> 4. How does DHCF exploit the relational dependencies across different types of user behaviors? (Q1)
>
> - Thank you for providing the feedback. In DHCF, we integrate cross-type contrastive learning tasks at both the node level and the graph level to leverage the relational dependencies across different types of user behaviors. At the node level, our contrastive learning approach aligns the learned node representations across diverse heterogeneous interaction types, facilitating the identification of shared characteristics across different behavior modes. Additionally, at the graph level, our contrastive learning aligns the representations for the entire graph, taking into account the cross-type commonalities in global graph structures.
>
> 5. How does the parameterized heterogeneous hypergraph architecture capture the complex and diverse interactions among users, items, and behaviors? (Q1)
>
> - Our parameterized hypergraph networks capture user-wise and item-wise relations from a global perspective by utilizing hypergraph-based global relation learning for each behavior type. This approach enables us to effectively model the heterogeneous interactions and fuse multi-type representations, thereby facilitating the modeling of relational dependencies between different entities.
>
> 6. How does DHCF disentangle the latent intent factors behind each behavior? (Q2)
>
> - DHCF leverages a parameterized hypergraph neural network to disentangle the latent intent factors that underlie each user behavior. In this architecture, the hyperedges in the hypergraph represent the latent user intents. Through iterative hypergraph structure learning, DHCF establishes connections between users/items and their associated hyperedges, effectively capturing users' frequent intents. This disentanglement process enables DHCF to model and differentiate the underlying intent factors that drive different user behaviors.
>
> 7. How does the behavior-wise contrastive learning paradigm facilitate adaptive data augmentation at both the node and graph levels? (Q2)
>
> - In DHCF, the adaptive data augmentation primarily targets the node level and involves personalized transformations tailored to different users and behavior types. Specifically, DHCF incorporates user-specific embedding transformations for their type-specific embeddings, which enables accommodating user-specific variations in behavior types. This approach allows DHCF to adaptively augment the data at a personalized level, enhancing the modeling of user behaviors with consideration for individual differences.

---

### Official Review · Reviewer_TWFu · 2023-11-01

**Soundness:** 3 good
**Presentation:** 4 excellent
**Contribution:** 3 good
**Rating:** 6
**Confidence:** 4

**Summary:**

This paper introduces a Disentangled Heterogeneous Collaborative Filtering (DHCF) for a recommendation system. Specifically, the model integrates a parameterized heterogeneous hypergraph network with a hierarchical contrastive learning paradigm, to capture the latent intent factors and the multi-behavior dependencies in an adaptive and self-supervised manner.

**Strengths:**

1. The task of recommendation with heterogeneous interactions is interesting and valuable.
2. The paper is written well and is easy to understand.
3. Extensive experiments have been conducted to validate the proposed model.

**Weaknesses:**

1. The methods proposed in the paper lack innovation significantly. Sections 3.1-3.5 follow very common design paradigms, and their method designs exhibit a certain degree of similarity to HCCF, ICL, and others. The paper should discuss the differences in the technical details between them.
2.  In section 3.6, there are two loss functions proposed for relationship learning but in reality, they belong to the same paradigm. The paper lacks sufficient theoretical justification for their validity.
3.  The font size in Figure 1 is too small.

**Questions:**

My major concern lies in the technical details. Many of the described methods bear a resemblance to existing approaches. It is crucial to clearly explain the distinctions and improvements made by DHCF in comparison to these existing methods. Additionally, please provide a more detailed explanation of the motivations behind these improvements.

---

> ### Author Response · Authors · 2023-11-20
>
> Responses to Reviewer TWFu
>
> Thank you for your valuable feedback. We would like to address your comments as follows:
>
> 1. Further discussion about the our encoder novlty. (W1 & Q)
>
> Our methodology comprises two primary components: the encoding part, which utilizes a heterogeneous hypergraph neural network, and the self-supervised learning part, employing heterogeneous contrastive learning. In order to address these concerns and highlight the technical novelty of our DHCF model, we provide the following justifications:
>
> - **Uniqueness of DHCF's Encoding Part**. While our DHCF model and existing works such as HCCF share some commonalities (e.g., the use of hypergraph neural networks), there are two significant technical differences in the design of our encoder: **heterogeneity** and **disentanglement**. Unlike existing hypergraph-based methods like HCCF, which primarily focus on homogeneous data, our DHCF model addresses the challenge of heterogeneity by considering interaction types and effectively preserving heterogeneity information in the hypergraph-based encoder. Furthermore, the adoption of a parameterized hypergraph neural network in our DHCF model facilitates efficient and learnable intent disentanglement. This disentanglement is vital not only during the encoding process but also in the subsequent self-supervised learning part, as discussed below. These technical distinctions highlight the unique contributions of our DHCF model, enabling it to handle heterogeneity and disentangle user intents more effectively compared to existing approaches.
>
> - **Novelty of Our Contrastive Learning and Its Relation to Encoder**. Our paper introduces a significant technical contribution in the form of the proposed disentangled heterogeneous contrastive learning method. This method incorporates several key components, including a parameter personalization module, a node-level heterogeneous self-discrimination task, and a graph-level contrastive learning task. These components leverage hypergraph-based global representation and shuffling-based negative sample construction techniques. By employing these designed techniques, our method greatly enhances the training of models for heterogeneous collaborative filtering. It achieves this by facilitating adaptive supervision enhancement while considering global intent disentanglement. Notably, our approach outperforms vanilla graph contrastive learning methods such as SGL, primarily due to the valuable heterogeneity and disentanglement information extracted by our designed encoding part.
>
>
> - **Theoretical Contribution**. Our paper makes a contribution by providing an in-depth theoretical analysis that highlights two important advantages of DHCF that distinguish it from existing works: i) hypergraph-based disentanglement enhances the adaptability of contrastive supervision signals, ii) graph-level contrastive learning injects global disentangled representations into node-level similarities. Thank you for your insightful feedback. We will further highlight the novelty and uniqueness of our encoding component in future versions.
>
> 2. Further theoretical justification of SSL loss terms (W2)
>
> - The two SSL loss functions employed in our study follow the contrastive learning paradigm. However, they differ in their specific focuses and objectives. The first SSL loss function concentrates on micro-scale learning, targeting the individual nodes within the graph. It enables fine-grained learning at the node level, capturing local heterogenous collaborative relationships among users and items. The second SSL loss function places emphasis on macro-scale learning, aiming to maximize learning for the overall graph structure. This macro-scale approach ensures that the model captures the global context and structural information within the multi-behavior data. Furthermore, since natural negative samples are lacking at the graph level, we introduce a shuffling-based negative sampling generation method, which effectively enlarges the technical similarity between the two loss functions.
>
> 3. Font size of Fig.1 is too small. (W3)
> - Thank you for your thoughtful suggestion. We will improve the clarity by updating the figure in later versions.

---

> > ### Comment · Reviewer_TWFu · 2023-11-21
> > **Thanks for the responses**
> >
> > Thank you for the responses, which have addressed my concerns to some extent. After carefully considering the comments from other reviewers, I have decided to withhold my score at this time.

---

### Meta-Review · Area_Chair_mxcT · 2023-12-28

**Metareview:**

The authors introduce "Disentangled Heterogenous Collaborative Filtering" (DHCF) to construct a recommendation system that: (1) can account for relational dependencies between heterogenous behavioral interactions (e.g., page view, add-to-favorite, add-to-cart, purchase) and (2) accounts for fine-grained latent factors associated with each behavior type. Methodologically, they adapt/modify several existing methods to: (1) disentangle intents and (2) develop a heterogenous hypergraph to capture diverse relationships between users and items (as determined by interaction type). Finally, they use multi-relation hierarchical contrastive learning approach (at the node and graph level) to optimize the model.

Consensus strengths identified by reviewers regarding this submission include:
- Accounting for heterogenous, but correlated, behavioral system interactions is of increased recent interest in the RecSys community and has strong potential commercial and practical interest.
- The proposed approach is sensible, shown to work well, and ostensibly has a path to further improvements.
- The paper is well-written, well-structured, and easy to understand (even if Figure 1 is somewhat impenetrable, especially on a printed paper version).
- The experiments are notably exhaustive and sufficiently validate the DHCF model.

Conversely, consensus limitations included:
- This work builds on several existing methods enumerated in Section 3 of the paper. The reviewers (and myself without additional reading) had difficulty determining which components were a direct translation, which required modifications, and what the precise modifications were. Ideally, more discussion would also motivate choosing the particular methods, if there are alternatives, etc. (conditioned on space limitations obviously). Some of this was addressed during rebuttal, but the reviewers continued to have concerns in this regard.
- Even after understanding the specific contributions, there was continued concern regarding the significance of the methodological contribution (i.e., while potentially dismissive, this can be cast as 'just' another hypergraph formulation that modifies the modeling to account for heterogeneity, but keeps the basic optimization procedure). This still makes for a potentially nice paper, but limits the scope of interest to RecSys.
- One reviewer found the robustness analysis unconvincing. In my reading, I had some difficulty understanding the details of the ablation study. While there are space constraints, additional discussion would be useful in understanding the authors' interpretation of these results.

My own additional (albeit slight) concern regarding this work is its level of interest and influence outside of the RecSys community as the results seem tailored to this particular (important) setting.

Overall, the reviewers agree that this submission addresses an important RecSys problem of current interest, accounting for heterogenous behavior signals. The approach paper is well-structured, well-written, and the empirical results are promising. The primary concerns were regarding a clearer description of the methodological novelty and experiments that more clearly disentangle the contribution of each component. Honestly, I think this is a nice paper, but likely borderline based on reviewer feedback and competition relative to papers that would have greater 'ML level' impact.

**Justification For Why Not Higher Score:**

The two points raised by the reviewers that I think are valid are: (1) better description of the precise contribution beyond identifying the problem and (2) additional discussion regarding the empirical results. I think the paper is borderline based on these, but might have leaned toward accepting if the contribution wasn't so focused on a specific RecSys problem (just that there may be a more suitable venue). I wouldn't object to it being accepted as it is a solid paper, but none of the reviewers (nor myself) felt it was necessary to accept.

**Justification For Why Not Lower Score:**

N/A

---

### Decision · Program_Chairs · 2024-01-16

Reject